# Prevalence of Congenital Disorders of Glycosylation in Childhood Epilepsy and Effects of Anti-Epileptic Drugs on the Transferrin Isoelectric Focusing Test

**DOI:** 10.3390/genes12081227

**Published:** 2021-08-10

**Authors:** Grace Silver, Shalini Bahl, Dawn Cordeiro, Abhinav Thakral, Taryn Athey, Saadet Mercimek-Andrews

**Affiliations:** 1Department of Pediatrics, Division of Clinical and Metabolic Genetics, The Hospital for Sick Children, Toronto, ON M5G 1X8, Canada; grace.silver@sickkids.ca (G.S.); shal.bahl@mail.utoronto.ca (S.B.); dawn.cordeiro@sickkids.ca (D.C.); abhinav.thakral@mail.utoronto.ca (A.T.); 2Department of Medical Genetics, University of Alberta, Stollery Children’s Hospital, Edmonton, AB T6G 2H7, Canada; Taryn.Athey@albertahealthservices.ca

**Keywords:** epilepsy, seizures, global developmental delay, congenital disorders of glycosylation, transferrin isoelectric focusing

## Abstract

Introduction: Childhood epilepsy is one of the most common neurological problems. The transferrin isoelectric focusing (TIEF) test is a screening test for congenital disorders of glycosylation (CDG). We identified abnormal TIEF test in children with epilepsy in our epilepsy genetics clinic. To determine if an abnormal TIEF test is associated with anti-epileptic medications or abnormal liver functions, we performed a retrospective cohort study. Methods: This study was performed between January 2012 and March 2020. Electronic patient charts were reviewed. Standard non-parametric statistical tests were applied using R statistical software. Fischer’s exact test was used for comparisons. Results: There were 206 patients. The TIEF test was abnormal in 11% (23 out of 206) of the patients. Nine patients were diagnosed with CDG: *PMM2*-CDG (*n* = 5), *ALG3*-CDG (*n* = 1), *ALG11*-CDG (*n* = 2), *SLC35A2*-CDG (*n* = 1). We report 51 different genetic diseases in 84 patients. Two groups, (1) abnormal TIEF test; (2) normal TIEF test, showed statistically significant differences for abnormal liver functions and for valproic acid treatment. Conclusion: The TIEF test guided CDG diagnosis in 2.9% of the patients. Due to the high prevalence of CDG (4.4%) in childhood epilepsy, the TIEF test might be included into the diagnostic investigations to allow earlier and cost-effective diagnosis.

## 1. Introduction

Childhood epilepsy is one of the most common neurological problems. The origin of epilepsy is known in about one-third of the children. There are numerous diagnostic investigations for childhood epilepsy. Some of these investigations are performed in pediatric, genetic, and neurology clinics to investigate the underlying etiologies for appropriate management. Metabolic investigations include plasma amino acids, acylcarnitine profile, total and free carnitines, homocysteine, transferrin isoelectric focusing (TIEF) tests, urine organic acids, and urine creatine and guanidinoacetate that may identify some of the treatable inherited metabolic disorders. Genetic investigations include microarray, a targeted next generation sequencing panel (TNGSP) for epilepsy, and exome sequencing (ES).

We previously reported the diagnostic yield of metabolic investigations (7%) in childhood epilepsy [1]. The diagnostic yield of TNGSP for epilepsy was 19%, and the diagnostic yield of ES was 37% in our recent study [2].

The TIEF test is applied as a screening test for congenital disorders of glycosylation (CDG) in patients with global developmental delay and epilepsy in our epilepsy genetics clinic. Glycosylation is a process where sugar groups are made, modified and added onto proteins. Several rare, genetic metabolic disorders produce defects in glycosylation, with CDG being an umbrella term for these inherited metabolic disorders. The first CDG was reported in 1984 by Jaeken et al. [3]. Since then, more than 130 different genetic defects were reported under the category of CDG. Due to the genetic heterogeneity of CDG, there are numerous phenotypes associated with specific genotypes. Most patients present with multisystemic disease with a significant neurological component, often in the form of developmental delay. The TIEF test, which measures transferrin isoforms as a screening test of CDG type I (CDG-I) and type II (CDG-II), has been used for more than 25 years. We recently reported six different types of CDG in 15 patients from our center. About 50% of those patients presented with seizures or epilepsy [4].

We identified an abnormal TIEF test in children with epilepsy in our epilepsy genetics clinic who were seen for diagnostic investigations. We performed a retrospective cohort study to determine: (1) if an abnormal TIEF test is associated with a specific seizure type; (2) if an abnormal TIEF test is associated with certain anti-epileptic medications; (3) if an abnormal TIEF test is associated with abnormal liver functions; and (4) if an abnormal TIEF test is associated with any type of genetic disorder. We reviewed the literature for associations between anti-epileptic medications, abnormal liver functions, and abnormal TIEF tests.

## 2. Materials and Methods

This retrospective cohort study was performed in a single pediatric epilepsy genetics clinic at an academic health center between January 2012 and March 2020. Inclusion criteria were: (1) non-syndromic epilepsy; (2) seen in this epilepsy genetics clinic for diagnostic investigations; (3) underwent TIEF testing. Exclusion criteria were patients who had no TIEF test AND patients who had no history of epilepsy or seizure. The Institutional Research Ethics Board at The Hospital for Sick Children approved the study (Approval #1000070163). Electronic patient charts were reviewed for clinical features, anti-epileptic medications, biochemical investigations, electrophysiological investigations, neuroimaging, and molecular genetic investigations including TNGSP, and ES. Information was entered into an Excel database (Microsoft Corp., Redmond, WA, USA). Liver function tests including AST, ALT, GGT, ALP, LDH, bilirubin, PT, and PTT, which were performed within one month of the TIEF test, were recorded. All anti-epileptic medications prior to and taken around the day(s) of the TIEF test were recorded. Patients’ and parents’ DNA samples were used for molecular genetic investigations according to the methods of clinical molecular genetic laboratories. The Alamut database was used for variant annotation for amino acid sequences, predictions of pathogenicity, and cross-species conservation of nucleotides. ACMG guidelines for the interpretation of variants were applied [5]. The Genome Aggregation Database was searched (gnomAD) (http://gnomad.broadinstitute.org/about, accessed between February–May 2021) for the allele frequency of variants in the general population [6].

Standard non-parametric statistical tests were used for within-group comparisons. All analyses were performed using R statistical software. Fischer’s exact test was used for comparisons. A two-tailed *p* value < 0.05 was considered statistically significant.

## 3. Results

There were 206 patients with childhood epilepsy fulfilling the inclusion criteria. All patients with abnormal TIEF tests are summarized for their clinical features, seizure history, anti-epileptic medications, liver function tests, and TIEF tests (patient and parents) in Table 1. We depicted the number of patients with normal and abnormal TIEF tests, their genetic diagnosis and type of molecular genetic investigation in Figure 1.

The TIEF test was abnormal in 11% (23 out of 206) of the patients. In 35% (8 out of 23) of those patients, CDG genetic diagnoses were confirmed by different genetic investigations. We used the current nomenclature for CDG, *gene*-CDG for all subgroups of CDG throughout the manuscript. There were eight patients with different types of CDG including *PMM2*-CDG (*n* = 5), *ALG3*-CDG (*n* = 1), *ALG11*-CDG (*n* = 1), and *SLC35A2*-CDG (*n* = 1). In 22% (5 out of 23) of the patients with an abnormal TIEF test, non-CDG genetic diagnoses were confirmed, including *EP300* disease (*n* = 1), *GRIN1* disease (*n* = 1), *HIVEP2* disease (*n* = 1), *KCNA2* disease (*n* = 1), and *KCNQ2* disease (*n* = 1). All variants are summarized in Appendix A. In the remaining 10 patients, only six of them had ES which was normal. Three patients had TNGSP for epilepsy which included only one CDG gene called *ALG13*. The false positive rate of the TIEF test was 5.3% (11 out of 206) in the study cohort. The false negative TIEF test rate was 0.5% (1 out of 206).

In the study cohort, CDG genetic diagnoses were confirmed in 4% of the patients (9 out of 206) by (1) an abnormal TIEF test and direct sequencing of *PMM2* or *ALG3* (*n* = 4); (2) an abnormal TIEF test and targeted next generation sequencing panel for CDG (*n* = 2); (3) an abnormal TIEF test and ES (*n* = 2); (4) a normal TIEF test and ES (*n* = 1). All variants are summarized in Appendix A.

One hundred and eight (52.4%) patients underwent ES, and its diagnostic yield was 40.7% (44 out of 108) in the study cohort. One hundred and sixty-six patients underwent TNGSP, and its diagnostic yield was 20.5% (34 out of 166) in the study cohort. One hundred and ninety-eight patients underwent microarray, and its diagnostic yield was 0.5% (1 out of 198 patients).

There were 51 different genetic diagnoses in 84 patients (48 previously published and 36 new patients) (Appendix A). All genetic diagnoses and the number of patients for each genetic disease are depicted in Figure 2. Interestingly more than half (44 out of 84) of the patients were diagnosed by ES. The remaining patients were diagnosed by various types of molecular genetic investigations including TNGSP for epilepsy (*n* = 29), for Aicardi–Goutières Syndrome (*n* = 1), for Cornelia de Lange syndrome (*n* = 1), for neuronal ceroid lipofuscinosis (*n* = 1), for pyruvate dehydrogenase complex (*n* = 1), for CDG (*n* = 2), and targeted Sanger sequencing of candidate genes (*n* = 5). One hundred and twenty-two (59.2%) patients had no genetic diagnosis and underwent ES (*n* = 64), and TNGSP (*n* = 52).

One hundred and two patients had at least one liver function test performed, including liver enzymes (AST and/or ALT *n* = 99), liver function tests for cholestasis (ALP and/or GGT and/or bilirubin *n* = 80), and/or INR (*n* = 24). Twenty-nine patients had elevated AST and/or ALT including with a normal (*n* = 24) TIEF test and an abnormal (*n* = 5) TIEF test. Eight patients had elevated ALP and/or GGT and/or bilirubin including with a normal (*n* = 3) TIEF test and an abnormal (*n* = 5) TIEF test. Eight patients had elevated INR including with a normal (*n* = 3) TIEF test and an abnormal (*n* = 2) TIEF test. There was no association between abnormal liver functions and abnormal TIEF tests (*p* value of 0.9904). There was no association between anti-epileptic medications and abnormal TIEF test (*p* value of 0.544). There was no association between abnormal liver functions and anti-epileptic medications (*p* value of 0.9959).

There were 93 variants in 50 different genes in 83 patients (one patient was diagnosed with 18qdel syndrome by microarray). Forty-eight patients were reported previously [2,4,7]. The ACMG variant classification of 43 patients were reported previously too [2,7]. In silico analysis and ACMG variant classification of the remaining 35 patients (additionally five previously published patients by Al Teneiji et al. [4]) are listed in Appendix A. There were 46 different variants from 40 patients (excluding patients with microarray abnormality), including 24 novel and 22 known variants. According to ACMG variant classification, 22 variants were pathogenic, 17 variants were likely pathogenic, and 7 variants were variants of uncertain significance. Segregation was confirmed in both parents in 30 patients and in one parent in four patients and in none of the parents in six patients.

We grouped patients into two groups: (1) abnormal TIEF test; (2) normal TIEF test. We compared both groups for their demographics, seizure types, anti-epileptic medications, and liver functions to see if there was any difference between both groups (Table 2). Interestingly, there were statistically significant differences between groups for liver functions including ALT, AST, and GGT, which was elevated in the group with abnormal TIEF tests. Additionally, there was a statistically significant difference between groups for valproic acid treatment with valproic acid being more commonly used in the group with abnormal TIEF tests.

We grouped patients into two groups, including: (1) with genetic diagnosis; (2) with no genetic diagnosis, and compared both groups for their demographics, seizure types, anti-epileptic medications, liver functions, and TIEF test results to see if there was any difference between groups (Table 3). There was a statistically significant difference for atonic seizures in the group with no genetic diagnosis. There was also a statistically significant difference for elevated asialotransferrin and disialotransferrin in the group with genetic diagnosis, which is likely associated with different types of CDG diagnosis in the group with genetic diagnosis.

## 4. Discussion

The prevalence of CDG was 4.4% in childhood epilepsy in our study. We report nine patients with four different CDG. We found a significant correlation between abnormal liver enzymes and abnormal TIEF test. We also found a significant correlation between valproic acid use and an abnormal TIEF test. Interestingly, the false positive rate of the TIEF test was 5.3% with a small percentage of false negative TIEF test rate (0.5%) in our study cohort. To the best of our knowledge, our study is the first study reporting the prevalence of CDG in childhood epilepsy and an abnormal TIEF test possibly related to anti-epileptic medications.

Hepatotoxicity associated with anti-epileptic medications is well known and has been summarized in review articles previously. Phenytoin, carbamazepine, valproic acid, and lamotrigine are commonly reported for their hepatoxic side-effects [8,9]. Valproic acid use is also associated with hyperammonemia and encephalopathy, toxic hepatitis, and Reye-like syndrome [8,9]. Carbamazepine and phenytoin-associated hepatotoxicity is characterized by asymptomatic transient elevations in liver enzymes occurring in up to 75% of patients [9]. Severe hepatocellular injury leading to liver failure has been reported in phenytoin, carbamazepine, and valproic acid use as the most severe spectrum of hepatotoxicity [9,10]. Most hepatotoxic reactions are idiosyncratic or immune hypersensitivity reactions [8,9]. Reactive arene oxide metabolites of phenytoin, carbamazepine, valproic acid, and lamotrigine are thought to be important in hepatotoxicity where they may cause oxidative injury and secondary immune response [8,9]. Valproic acid specifically is thought to cause hepatotoxicity through the inhibition of mitochondrial β-oxidation by reactive metabolites [8,9]. Frequency of valproic acid hepatoxicity is up to 1 in 800 in children under the age of two years [11]. In our study, one-quarter of the patients had abnormal liver functions. There was no significant association between abnormal liver functions and anti-epileptic medications (*p* value 0.9959) and between anti-epileptic medications and abnormal TIEF test (*p* value 0.544).

Abnormal TIEF test have been associated with liver disease or inherited metabolic disorders affecting liver functions, such as hereditary fructose intolerance and galactosemia [12,13]. In a recent study, 1546 individuals underwent TIEF test [14]. An abnormal TIEF test was identified in 3% (51 out of 1546) of those individuals. There were follow-up investigations in 14 of those patients. Only four patients received a confirmed diagnosis of CDG: *PMM2*-CDG (*n* = 2), *MPDU1*-CDG (*n* = 1), and *SLC35A2*-CDG (*n* = 1), whereas in 10 patients, an abnormal TIEF test was secondary to other inherited metabolic disorders: galactosemia (*n* = 4), hereditary fructose intolerance (*n* = 2), and peroxisomal disease (*n* = 2). Individuals with liver disease had a 4.6 times higher likelihood of abnormal TIEF test results [14]. In another study, 554 children with a suspected CDG underwent TIEF test and nine of them had an abnormal TIEF test. Four patients had *PMM2*-CDG, and two patients had *ALG2*-CDG [15]. In our study, 4.4% (9 out of 206) of the patients had genetically confirmed CDG and most of those patients had an abnormal TIEF test to suggest CDG. We identified other inherited metabolic disorders including pyridoxine-dependent epilepsy (PDE) due to biallelic variants in *ALDH7A1* [4], *PYCR2* disease [16], *EARS2* disease [17], *PARS2* disease [17], *CLN7* disease, and glucose transporter 1 deficiency syndrome with or without liver dysfunction in our study cohort. We did not identify any patients with galactosemia or hereditary fructose intolerance in our study cohort. We think that the TIEF test might be included into the diagnostic investigations as screening test of CDG.

In a recent study, glycosylation was studied in 981 patients with adult liver disease to investigate if there was any difference between primary and secondary glycosylation defects in liver disease. An abnormal TIEF test was identified in 26% of the patients, including increased a percentage of trisialotransferrin (70%), monosialotransferrin (17%), disialotransferrin (4%) and mixed transferrin isoforms (11%) [18]. None of the patients had a CDG diagnosis [18]. There is no information for neurological phenotypes of those patients and trisiaolotransferrin is not one of the typical CDG type I or II patterns. The study had no information regarding the underlying causes of liver disease [18]. In another study, 19 pediatric liver disease patients underwent a TIEF test [19]; 17 of them had elevated asialotransferrin and monosialotransferrin, and 13 of them had elevated disialotransferrin. Eight patients had follow-up liver function tests and TIEF test were normalized in all eight patients during follow-up. Two patients had inherited metabolic disorders: citrin deficiency (*n* = 1) and deoxyguanosine kinase deficiency (*n* = 1) [19]. Our patient population is unique, as we did not select patients with liver disease, but included all patients with epilepsy who had TIEF test for diagnostic investigations.

Transferrin is synthesized by the endoplasmic reticulum and golgi apparatus in the liver. Hepatocyte injury results in endoplasmic reticulum damage and impaired excretory function of glycosylated proteins [19]. Changes in the TIEF test profile in liver disease represent alterations occurring during biosynthesis of transferrin by the endoplasmic reticulum (ER) and golgi apparatus [20]. Several changes have been reported in individuals with liver disease: (1) chronic alcohol abuse is known to alter the glycosylation pattern with an increase in asialotransferrin and disialotransferrin [21]; (2) polymorphisms in transferrin B2 and C2 may result in abnormal TIEF test characterized by elevated pentasialotransferrin and trisialotransferrin [21]; (3) fucosyltransferase activity and fucosylation are increased in alcoholic liver disease, cholestatic liver disease, and hepatocellular carcinoma [20]; (4) sialyltransferase, galactosyltransferase, and mannosyltransferase activity changes have been reported in alcoholic liver disease [18,19]. It has been shown that valproic acid impaired glycosylation and the secretion of acid phosphatase in fission yeast model. The authors speculated that this could contribute to adverse effects of valproic acid in humans [22]. Taken together, liver injury affects glycosylation. There might be a causative link for the abnormal glycosylation associated with anti-epileptic treatment and their effects on the cellular and synthetic liver functions in childhood epilepsy.

When proteins and lipids are combined with sugars, advanced glycation end-products (AGE) are generated which are implicated to contribute to disease progression. Receptor for AGE (RAGE) is found in different cells. When RAGE binds to AGE, nuclear factor kappa B (NFkB) is activated and releases inflammatory mediators, which contributes to inflammation, intracellular damage and apoptosis [23,24,25]. In brain biopsy samples of individuals with temporal lobe epilepsy, RAGE was upregulated in astrocytes, neurons, and microvessels and was reported to contribute to seizures [23]. In seizing mice, RAGE was upregulated, while RAGE knockout mice had reduced seizure activity [23]. The presence of G28S polymorphism in RAGE was significantly more common in patients with drug-resistant epilepsy in a study [25]. In individuals with epilepsy, increased blood sugar levels have been reported, which may be associated with AGE and poor seizure control in childhood epilepsy. While a mechanism for abnormal glycosylation associated with epilepsy and antiepileptic treatment is unknown, we propose a possible link with RAGE.

Post-translational modification is a protein modification process to improve structure and function of proteins after protein synthesis. Interestingly EP300, GRIN1, KCNA2, and KCNQ2 proteins undergo post-translational modification either by amino acid modifications and N-linked glycosylation (KCNA2 and GRIN1 proteins) or ubiquitylation and O-glycosylation (KCNQ2 and EP300 proteins). It is likely that pathogenic or likely pathogenic variants in these genes may affect the post-translational modification of these proteins via N- or O-glycosylation and cause abnormal TIEF test. Further research is necessary to understand better the implications of the variants on the post-translational modification of proteins.

Limitations of our study include: (1) a retrospective cohort study that reports patients with epilepsy seen in a single epilepsy genetics clinic for diagnostic investigations; (2) exome sequencing was not applied to all patients; (3) the study was initiated in the metabolic unit; this has likely contributed to the higher number of patients with CDG; (4) there might have been other CDG patients at our institution who were diagnosed during our study period that were not included into this study, as they were not known to the authors; (5) we have no patient database to perform an institution-wide retrospective study for all patients with epilepsy who underwent transferrin isoelectric focusing; (6) we do not know the diagnostic yield of the TIEF test at our institution. Despite these limitations, we think that our study results provide a diagnostic yield of TIEF test and a high prevalence of CDG in childhood epilepsy.

In summary, interestingly, the TIEF test guided CDG diagnosis in 2.9% (6 out of 206) of the patients and prevented the use of untargeted molecular genetic investigations such as ES. Due to the high prevalence of CDG in childhood epilepsy, the TIEF test might be included in the diagnostic investigations. Application of the TIEF test in childhood epilepsy may allow physicians to reach a diagnosis in a shorter period of time and be cost effective.

## Figures and Tables

**Figure 1 genes-12-01227-f001:**
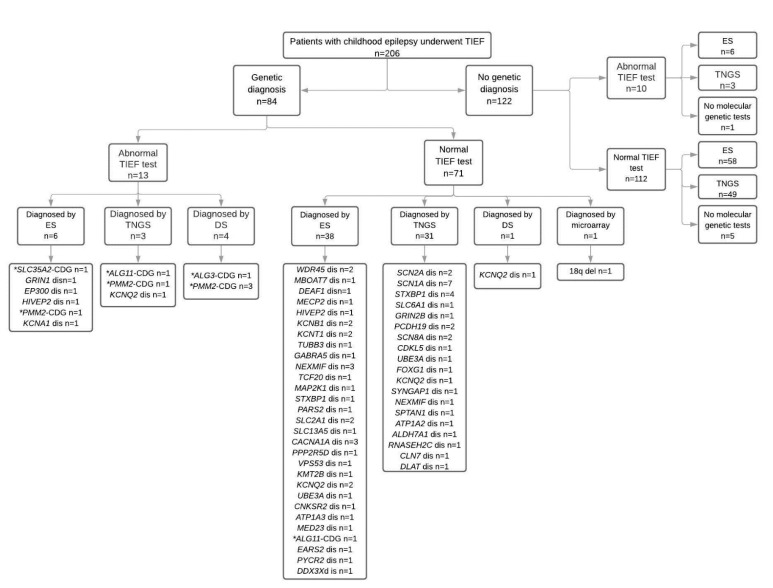
Number of patients with normal and abnormal TIEF tests, their genetic diagnosis and type of molecular genetic investigation are depicted in Figure 1. * Statistically significant.

**Figure 2 genes-12-01227-f002:**
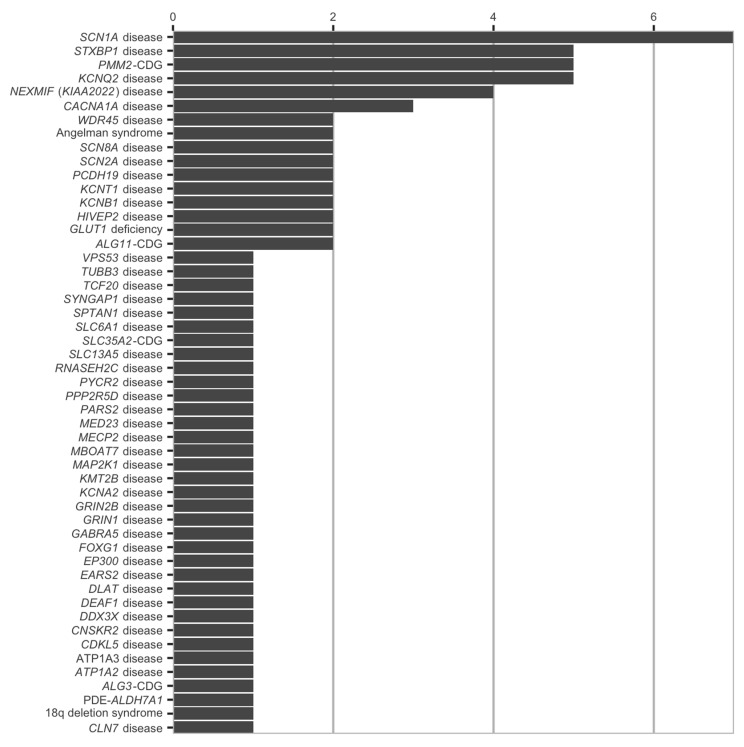
All genetic diagnoses and the number of patients for each genetic disease are depicted in Figure 2.

**Table 1 genes-12-01227-t001:** All patients with abnormal transferrin isoelectric focusing tests, their clinical features, liver functions, and transferrin isoelectric focusing results are summarized.

Patient Number/ Study ID/Sex/Current Age (Reference)	Diagnosis (Genetic or None) (Age of Diagnosis)	Seizure Age of Onset/Seizure Types	Other Clinical Features	Anti-Epileptic Medications Used	Anti-Epileptic Medications at the Time of TIEF Test	Liver Functions AST/ALT/INR/GGT/ ALP/Direct Bil	TIEF Test	Parental TIEF
1/015/F/2 yr(s)	*SLC35A2*-CDG (8 mo(s)) by ES	11 mo(s)/IS	GDD, FTT, dysmorphic features (hypertelorism, low set posteriorly rotated ears, prominent forehead, upslanting palpebral fissures, short nose with upturned nose tip)	TPM, PRED, VGB	None	↑/N/NA/N/NA/NA	↑asialo, mono, di, tri ↓tetra, penta	NA
2/031/F/10 yr(s) [1]	*ALG11*-CDG (5 yr(s)) by TNGSP for CDG (37 genes)	2 mo(s)/GTCS, GTS, IS	GDD, dystonia, microcephaly, dysmorphic features (mild frontal bossing, broad and tubular nose with new onset of milia, retrognathia, small down-turned mouth, chubby cheeks)	VGB, PRED, ACTH, TPM, LVT, CLB, LOR	VGB	N/N/NA/N/↓/N	↑disialo	NA
3/215/F/12 yr(s) [1]	*ALG3*-CDG (2 yr(s)) by TIEF & DS	Day 20/GTCS, GTS	GDD, ataxia, spasticity, dysmorphic features (plagiocephaly, micrognathia, tubular nose)	PB	PB	N/N/N/N/NA/N	↑asialo, ↑disialo	NA
4/210/M/19 yr(s) [1]	*PMM2*-CDG (21 mo(s)) by TIEF & DS	1 yr(s)/GTCS	GDD, visual problems	None	None	↑/↑/NA/N/NA/NA	↑asialo, ↑disialo	NA
5/211/M/18 yr(s)	*PMM2*-CDG (16 yr(s)) by ES	3.5 yr(s)/GTCS, MS, CPS	GDD, ataxia	CBZ, OXC	None	N/N/N/N/NA/NA	↑asialo, disialo ↓tetra	NA
6/222/F/7 yr(s) [1]	*PMM2*-CDG (15 mo(s)) by TNGSP for CDG (67 genes)	2 mo(s)/GTCS, GTS, focal	GDD, respiratory distress, cardiac abnormalities	PB, LOR	None	N/NA/NA/NA/N/N	↑disialo ↓trisialo, ↓tetrasialo	NA
7/224/F/4 yr(s) [1]	*PMM2*-CDG (4 mo(s)) by TIEF & DS	Day 12/GTS	GDD, FTT	None	None	↑/↑/N/NA/NA/NA	↑disialo ↓tetrasialo	NA
8/230/M/3 yr(s)	*PMM2*-CDG (3 yr(s)) by TIEF & DS	18 mo(s)/GTCS	GDD, FTT, spasticity, microcephaly, dysmorphic features (inverted nipples, low-set ears)	None	None	↑/↑/N/↑/N/N	↑asialo, disialo ↓trisialo, tetrasialo	NA
9/057/F/15 yr(s) [2]	*KCNA2* disease (10 yr(s)) by ES	8 mo(s)/GTCS, GTS, MS, AbS	GDD, ataxia	PB, LVT, VPA, OXC	LVT, VPA	NA/NA/NA/NA/NA/NA	↑trisialo	Pat N Mat N
10/090/F/7 yr(s) [2]	*GRIN1* disease (7 yr(s)) by ES	Day 7/GTCS, AbS	GDD, spastic diplegia	PB, CBZ, VPA, CZP, LOR	VPA, CZP	↑/NA/NA/N/NA/NA	↑trisialo	Mat N
11/102/F/17 yr(s) [2]	*EP300* disease (11 yr(s)) by ES	8 yr(s)/AbS	GDD, ASD, dysmorphic features (triangular shaped face with prominent eyebrows, thin upper lip, narrow high arched palate, narrow forehead, posteriorly rotated ears, retrognathia, prominent frontal incisors)	VPA	VPA	N/N/NA/N/NA/NA	↑ trisialo ↓tetrasialo	Mat N
12/193/F/11 yr(s)	*HIVEP2* disease (6 yr(s)) by ES	20 mo(s)/GTCS, MS, AbS	GDD, ADHD	LVT, VPA	VPA, LVT	N/N/NA/NA/↓/N	Tetrasialo doublet	NA
13/197/M/13 yr(s) [2]	*KCNQ2* disease (7 yr(s)) by TNGSP for epilepsy (70 genes)	Day 1/GTCS, GTS	GDD, dysmorphic features (thick eyebrows, flat nasal bridge, prominent philtral groove, malar hypoplasia)	PHT, PB, TPM, LOR, MID, CLB	PB, TPM, CLB	NA/NA/NA/NA/NA/NA	↑trisialo	NA
14/016/M/3 yr(s)	None (ES negative)	5 mo(s)/IS	GDD	MID, VGB, TPM	None	↑/↑/N/N/↑/N	↑ asialo, disialo ↓tetrasialo	Pat N Mat N
15/021/M/4 yr(s)	None (TNGSP for epilepsy, 127 genes)	10 mo(s)/GTS, IS, MS, AbS, AS	GDD	VPA, LOR, GBP, CBD, CZP, TPM, VGB, ACTH	VPA, LOR, GBP	N/N/NA/↑/N/NA	↑trisialo	N/A
16/050/F/13 yr(s)	None (TNGSP for epilepsy, 87 genes)	2 yr(s)/GTCS, GTS, MS	GDD, ADHD	LVT, VPA, LOR	VPA	NA/NA/NA/NA/NA/NA	Tetrasialo doublet	N/A
17/059/M/7 yr(s)	None (ES negative)	2 yr(s)/GTCS, MS, AS	GDD	LOR, LVT, CBZ, VPA, CLB, TPM, FOS	CLB, TPM, VPA, LOR	N/N/↑/N/↓/NA	↑trisialo	Pat N Mat N
18/064/M/16 yr(s)	None (ES negative)	18 mo(s)/GTCS, MS, AbS	ASD	VPA, CBZ, ESM, LMT, LVT, TPM, CLB, RUF	VPA	NA/NA/NA/NA/NA/NA	↑trisialo	Pat N Mat N
19/066/M/16 yr(s)	None (microarray)	2.5 yr(s)/MS, AbS	Tremor, ADHD, ASD, temper dysregulation disorder, aggressive behaviour	VPA, CBZ, CLB, ESM, DZP	VPA	NA/NA/NA/NA/NA/NA	↑trisialo ↓tetrasialo	N/A
20/097/F/15 yr(s)	None (TNGSP for epilepsy, 87 genes)	6 yr(s)/GTCS, MS, CPS	ADHD	VPA, LMT, TPM, ESM	VPA, LMT	NA/NA/N/NA/NA/NA	↑trisialo	Pat:↓asialo, disialo, ↓tetrasialo Mat N
21/124/M/21 yr(s)	None (ES negative)	2 yr(s)/GTCS, AbS	Mild intellectual disability	ESM, VPA, LMT, LOR	ESM, VPA, LMT, LOR	NA/NA/NA/NA/NA/NA	↑trisialo	N/A
22/198/M/8 yr(s)	None (ES negative)	4 mo(s)/IS, MS	GDD	VGB, ACTH, VPA, CBD, CLB, LMT	VPA, CBD	N/N/NA/NA/N/NA	↑trisialo	N/A
23/208/M/9 yr(s)	None (ES negative)	4 yr(s)/AS	GDD, ASD, self-mutilation	LVT, CZP, LOR	None	NA/NA/NA/NA/NA/NA	Tetrasialo doublet	N/A

**Abbreviations:** AbS = absence seizure; AS = atonic seizure; ASD = autism spectrum disorder; CBD = cannabidiol; CBZ = carbamazepine; CLB = clobazam; CPS = complex partial seizure; CZP = clonazepam; DS = direct sequencing; DZP = diazepam; ES = exome sequencing; ESM = ethosuximide; FOS = fosphenytoin; FTT = failure to thrive; GBP = gabapentin; GDD = global developmental delay; GTCS = generalized tonic clonic seizure; GTS = generalized tonic seizure; IS = infantile spasms; LMT = lamotrigine; LOR =lorazepam; LVT = levetiracetam; MID = midazolam; MS = myoclonic seizure; OXC = oxcarbazepine; PB = phenobarbital; PHT = phenytoin; PRED = prednisone; RUF = rufinamide; TIEF = transferrin isoelectric focusing; TNGS =targeted next generation sequencing; TPM = topiramate; VGB = vigabatrin; VPA = valproic acid; N = normal; NA = not available; ↑ = increased; ↓ = decreased; Pat = paternal; Mat = maternal.

**Table 2 genes-12-01227-t002:** Comparison of patients with abnormal or normal transferrin isoelectric focusing for their demographics, seizure types, anti-epileptic medications, and liver functions.

	With Abnormal TIEF (*n* = 23)		With Normal TIEF (*n* = 183)		*p*-Value (Fisher Exact Test)
Median Age at Diagnosis (Months)	60		60	
Median Age at Onset (Months)	18		18	
	*n*	%	*n*	%
Sex (=Male)	12	52.17	90	49.18
**Liver function tests**					
AST	6	26.09	19	10.38	0.04164 *
ALT	5	21.74	11	6.01	0.02111 *
GGT	5	21.74	2	1.09	0.0002055 *
ALP	0	0	1	0.55	1
Bilirubin	0	0	2	1.09	1
INR	2	8.70	3	1.64	0.09693
**Anti-epileptic medications**					
Topiramate	4	17.39	24	13.11	0.526830097
Phenobarbitone	3	13.04	39	21.31	0.425122663
Clonazepam	1	4.35	8	4.37	1
Carbamazepine	0	0	11	6.01	0.615961230
Clobazam	3	13.04	54	29.51	0.136747826
Lorazepam	4	17.39	26	14.21	0.752853085
Valproic acid	12	52.17	38	20.77	0.003085312 *
Oxcarbazepine	0	0	8	4.37	0.601562786
CBD oil	1	4.35	7	3.83	1
Gabapentine	1	4.35	0	0	0.111650485
Diazepam	0	0	2	1.09	1
Vigabatrin	1	4.35	4	2.19	0.450170633
Ethosuximide	1	4.35	9	4.92	1
ACTH	0	0	3	1.64	1
Acetazolamide	0	0	1	0.55	1
Rufinamide	0	0	2	1.09	1
Perampenil	0	0	1	0.55	1
Midazolam	0	0	3	1.64	1
Phenytoin	0	0	4	2.19	1
Stiripentol	0	0	1	0.55	1
Lacosamide	0	0	1	0.55	1
**Types of seizures**					
Generalized seizures	17	73.91	149	81.42	0.4047
Partial seizures	3	13.04	57	31.15	0.08905
Infantile spasms	5	21.74	34	18.58	0.7776
Absence seizures	8	34.78	67	36.61	1
Atonic seizures	3	13.04	50	27.32	0.205
Myoclonic seizures	10	43.48	63	34.43	0.4883

* Statistically significant.

**Table 3 genes-12-01227-t003:** Comparison of patients with and without a genetic diagnosis for their demographics, seizure types, anti-epileptic medications, transferrin isoelectric focusing and liver functions.

Clinical Features & Results	With Genetic Diagnosis (*n* = 84)		Without Genetic Diagnosis (*n* = 122)		*p*-Value (Fisher Exact Test)
Median Age at Diagnosis (Months)	60		60	
Median Age at Onset (Months)	18		18	
	N	%	N	%
Sex (=Male)	33	39.29	69	56.56
**Liver function tests**					
AST	15	17.86	10	8.20	0.04985895
ALT	8	9.52	8	6.56	0.44088861
GGT	5	5.95	2	1.64	0.12384944
ALP	0	0	1	0.82	1
Bilirubin	1	1.19	1	0.82	1
INR	2	2.38	3	2.46	1
**Anti-epileptic medications**					
Levetiracetam	32	38.10	31	25.41	0.06483421
Topiramate	9	10.71	19	15.57	0.40901063
Phenobarbitone	22	26.19	20	16.39	0.11280733
Clonazepam	4	4.76	5	4.10	1
Carbamazepine	5	5.95	6	4.92	1
Clobazam	20	23.81	37	30.33	0.34373283
Lorazepam	14	16.68	16	13.11	0.54802867
Valproate	20	23.81	30	24.59	1
Oxcarbazepine	3	3.57	5	4.10	1
CBD oil	3	3.57	5	4.10	1
Gabapentin	0	0	1	0.82	1
Diazepam	1	1.19	1	0.82	1
Vigabatrin	1	1.19	4	3.28	0.65038894
Ethosuximide	1	1.19	9	7.38	0.05042569
ACTH	0	0	3	2.46	0.27198674
Acetazolamide	1	1.19	0	0	0.40776699
Rufinamide	0	0	2	1.64	0.51465783
Perampenil	0	0	1	0.82	1
Midazolam	1	1.19	2	1.64	1
Phenytoin	2	2.38	2	1.64	1
Stiripentol	1	1.19	0	0	0.40776699
Lacosamide	1	1.19	0	0	0.40776699
**Type of Seizures**					
Generalized Seizures	69	82.14	97	79.51	0.72135379
Partial Seizures	26	30.95	34	27.87	0.64291817
Infantile Spasms	11	13.10	28	22.95	0.10274649
Absence Seizures	24	28.57	51	41.80	0.05687798
Atonic Seizures	15	17.86	38	31.15	0.03568696 *
Myoclonic Seizures	26	30.95	47	38.52	0.30083751
**TIEF test**					
Asialotransferrin	5	5.95	1	0.82	0.042312860 *
Monosialotransferrin	1	1.19	0	0	0.407766990
Disialotransferrin	7	8.33	1	0.82	0.008505866 *
Trisialotransferrin	5	5.95	7	5.74	1
Tetrasialotransferrin	7	8.33	4	3.28	0.126938204
Pentasialotransferrin	0	0	1	0.82	1

* Statistically significant.

## Data Availability

Not applicable.

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
