# Peer review of "Prevalence of Congenital Disorders of Glycosylation in Childhood Epilepsy and Effects of Anti-Epileptic Drugs on the Transferrin Isoelectric Focusing Test"

_genes, 2021, doi:10.3390/genes12081227_

Round 1
Reviewer 1 Report
In a retrospective study, a cohort of paediatric patients with epilepsy a diagnostic value was studied for a screening on congenital disorders of glycosylation using isoelectric focusing of transferrin (TIEF). Important aspect of the study was to include a confounding effect of anticonvulsant therapy, potentially interfering with TIEF by their hepatotoxicity. The study group was well described on the molecular level. Study participants were already investigated by exome sequencing, targeted NGS panels or genomic microarray. However, I have some concerns to this manuscript.
- A declarative title suggest that diagnostic value of TIEF was studied in patients with congenital epilepsy. The cohort would be much better described by congenital neurological abnormalities with seizures. Some of these diagnosed syndromes have delayed psychomotor development, abnormal muscular tone, ataxia, microcephaly and many other signs, with epilepsy as a non-mandatory constituent of clinical synopsis,
- Of importance for practical paediatrics, the study subjects could be stratified as syndromic or non-syndromic children with epilepsy. One may easily guess, that the patient with EP300 mutation had Rubinstein-Taybi, or 18q deletion syndrome had accompanying dysmorphic features or organs malformations. Were among these diagnosed of CDG some syndromic cases or nonsydromic ones, were dysmorphic features present in the other 10 TIEF positive but without definite diagnosis?
- Results focused on detailed description of abnormal TIEF results. There was a lot of false positives TIEF results in the cohort. In children without established mutation, it seemed not to be explained by the potential hepatotoxicity of the drugs. These 10 subjects could have syndromes, which mutation was not captured by TNGS (n=3) or exome sequencing (n=6).
- Authors could discuss why potassium channels mutations; table 1, KCNA2, KCNQ2 or NMDA receptor (GRIN1) had abnorman TIEF test.
Author Response
Response to Reviewer 1
- A declarative title suggests that diagnostic value of TIEF was studied in patients with congenital epilepsy. The cohort would be much better described by congenital neurological abnormalities with seizures. Some of these diagnosed syndromes have delayed psychomotor development, abnormal muscular tone, ataxia, microcephaly and many other signs, with epilepsy as a non-mandatory constituent of clinical synopsis.
Answer: Thanks very much for this great question. We updated our title. We also added dysmorphic features into our Table 1 and Supplemental table 1 for the patients with dysmorphic features. We highlighted all changes in yellow. We hope that our responses are satisfactory to Reviewer 1.
- Of importance for practical paediatrics, the study subjects could be stratified as syndromic or non-syndromic children with epilepsy. One may easily guess, that the patient with EP300 mutation had Rubinstein-Taybi, or 18q deletion syndrome had accompanying dysmorphic features or organs malformations. Were among these diagnosed of CDG some syndromic cases or nonsydromic ones, were dysmorphic features present in the other 10 TIEF positive but without definite diagnosis?
Answer: Thanks very much for this great question. We added dysmorphic features into our Table 1 and Supplemental table 1 for the patients with dysmorphic features. We highlighted all changes in yellow. Our patient cohort was non-dysmorphic epilepsy which is also included into inclusion criteria. We hope that our responses are satisfactory to Reviewer 1.
- Results focused on detailed description of abnormal TIEF results. There was a lot of false positives TIEF results in the cohort. In children without established mutation, it seemed not to be explained by the potential hepatotoxicity of the drugs. These 10 subjects could have syndromes, which mutation was not captured by TNGS (n=3) or exome sequencing (n=6).
Answer: Thanks very much for this very interesting question. Our patient population was non-syndromic epilepsy patients combining patients referred to epilepsy genetics clinic. All syndromic patients were seen in the clinical genetic clinics for syndrome diagnosis, but not in our clinic. None of the patients were diagnosed by their dysmorphic features in our study cohort. We agree with the Reviewer 1 that the undiagnosed patients can have syndromes. We hope that our responses are satisfactory to Reviewer 1.
- Authors could discuss why potassium channels mutations; table 1, KCNA2, KCNQ2 or NMDA receptor (GRIN1) had abnormal TIEF test.
Answer: Thanks very much for this very interesting question. We included this information into the discussion and highlighted yellow. We hope that our responses are satisfactory to Reviewer 1.

Reviewer 2 Report
The article should have a minor revision that should focus on following points:
- The Authors should provide explanation for all medical abbreviations they use. The usage of abbreviations, such as, for example, PMM2-CDG or ALG3-CDG, without a clear explanation makes the text poorly comprehensible for non-specialist Readers.
- The number of patients with abnormal TIEF (n=23) is very different than the number of patients with normal TIEF (n=183). Was this fact taken into consideration while doing the statistical analysis ?
- I am not sure about the final conclusion that the TIEF test should be included in the diagnostic insvestigation. Is the fact that the test guided CDG diagnosis in 6 out of 206 patients enough to consider such a test as a recommended diagnostic tool ?
Author Response
Response to Reviewer 2
- The Authors should provide explanation for all medical abbreviations they use. The usage of abbreviations, such as, for example, PMM2-CDG or ALG3-CDG, without a clear explanation makes the text poorly comprehensible for non-specialist Readers.
Answer: Thanks very much for this question and our apologies for not clarifying this earlier. We used the CDG classification nomenclator and explained this in the manuscript (in the result section) and highlighted yellow. We hope that our responses are satisfactory to Reviewer 2.
- The number of patients with abnormal TIEF (n=23) is very different than the number of patients with normal TIEF (n=183). Was this fact taken into consideration while doing the statistical analysis?
Answer: Thanks very much for this great question. Yes, we applied appropriate statistical analysis for the different number of patients within groups. Shalini Bahl applied all statistical analysis taking into account this point and listed all statistical methods she used under statistical analysis in the methods section. We hope that our responses are satisfactory to Reviewer 2.
- I am not sure about the final conclusion that the TIEF test should be included in the diagnostic insvestigation. Is the fact that the test guided CDG diagnosis in 6 out of 206 patients enough to consider such a test as a recommended diagnostic tool?
Answer: Thanks very much for this great question. We agree with the Reviewer 1. We changed “should” to “might” in the abstract and in the discussion. We highlighted these changes yellow. We hope that our responses are satisfactory to Reviewer 2.
